# Evaluating the integration of Esper complex event processing engine and message brokers

Guadalupe Ortiz[1], Adrian Bazan-Muñoz[1], Winfried Lamersdorf[2] and Alfonso Garcia-de-Prado[3]

[1] UCASE Software Engineering Research Group, Department of Computer Science and Engineering, University of Cádiz, Puerto Real, Spain
[2] VSYS Distributed Systems Research Group, Department of Computer Science, University of Hamburg, Hamburg, Germany
[3] Software Engineering Research Group, Department of Computer Architecture and Technology, University of Cádiz, Puerto Real, Spain



Corresponding author
Guadalupe Ortiz,
guadalupe.ortiz@uca.es

## ABSTRACT

The great advance and affordability of technologies, communications and sensor technology has led to the generation of large amounts of data in the field of the Internet of Things and smart environments, as well as a great demand for smart applications and services adapted to the specific needs of each individual. This has entailed the need for systems capable of receiving, routing and processing large amounts of data to detect situations of interest with low latency, but despite the many existing works in recent years, studying highly scalable and low latency data processing systems is still necessary. In this area, the efficiency of complex event processing (CEP) technology is of particular significance and has been used in a variety of application scenarios. However, in most of these scenarios there is no performance evaluation to show how the system performs under various loads and therefore the developer is challenged to develop such CEP-based systems in new scenarios without knowing how the system will be able to handle different input data rates and address scalability and fault tolerance. This article aims to fill this gap by providing an evaluation of the various versions of one of the most reputable CEP engines—Esper CEP, as well as its integration with two renowned messaging brokers for data ingestion—RabbitMQ and Apache Kafka. For this purpose, we defined a benchmark with a series of event patterns with some of the most representative operators of the Esper CEP engine and we performed a series of tests with an increasing rate of input data to the system. We did this for three alternative software architectures: integrating open-source Esper and RabbitMQ, integrating one instance of Esper enterprise edition with Apache Kafka, and integrating two distributed instances of Esper enterprise edition with Apache Kafka. We measured the usage of CPU, RAM memory, latency and throughput time, looking for the data input rate with which the system overloads for each event pattern and we compared the results of the three proposed architectures. The results have shown a very low CPU consumption for all implementation options and input data rates; a balanced memory usage, quite similar among the three architectures, up to an input rate of 10,000 or 15,000 events per second, depending on the architecture and event pattern, and a quite efficient response time up to 10,000 or 15,000 events per second, depending on the architecture and event pattern. Based on a more exhaustive analysis of results, we have concluded that the different options offered by Esper for

CEP provide very efficient solutions for real-time data processing, although each with its limitations in terms of brokers to be used for data integration, scalability, and fault tolerance; a number of suggestions have been drawn out for the developer to take as a basis for choosing which CEP engine and which messaging broker to use for the implementation depending on the of the system in question.

# INTRODUCTION

Thanks to the enormous advances on the Internet of Things (IoT) and the emergence of smart scenarios in everyday life, large amounts of data are currently generated from multiple sources and are expected to be processed to gain a better understanding of the domain in question and make improved decisions in smart environments. Thus, there is a growing body of work seeking efficiency in real time and intelligent processing of massive amounts of streaming data, with a special emphasis on IoT data processing and smart scenarios (*Skarlat & Schulte, 2021*; *Ponce & Abdulrazak, 2022*). Despite the many existing works in recent years, the need to study systems that can perform the processing of large amounts of data in a scalable way (*Rahmani, Babaei & Souri, 2021*) and with low latency (*Bhatt & Thakkar, 2021*) is still highlighted as an open issue (*Babar et al., 2022*). In particular multiple frameworks for big data analysis in the IoT face numerous challenges such as the high volume of data and their heterogeneity, as well as processing time, among others.

In *Rahmani, Babaei & Souri (2021)* it is stated that complex event processing (CEP) has become a successful technology for streaming data processing and their integration with message brokers has provided a fundamental part of IoT and smart environments systems, since it facilitates the integration of multiple sources for their joint real-time processing, as supported by multiple publications (*Mayer, Koldehofe & Rothermel, 2015*; *Akbar et al., 2017*; *Garcia-de-Prado, Ortiz & Boubeta-Puig, 2017*). This use of CEP for the IoT is not only in the cloud, but also at levels closer to the device, such as the fog or the edge (*Mondragón-Ruiz et al., 2021*). Although high performance of streaming data processing is achieved with CEP and solutions for data heterogeneity have also been proposed with this technology (*Corral-Plaza et al., 2021*, p.; *Rath, Mandal & Sarkar, 2023*), we can sometimes encounter scenarios that require higher performance, not forgetting the unpredictable growth that is being experienced both in the quantity and speed of the data generated by the improvement in cyber-physical devices, as well as communications.

In this context, the developer may face limitations in terms of scalability and/or performance of the system once it has been implemented, leading to the need to evolve to other technologies or to replace the implemented software architecture with another one. For this reason, it is essential to have a reference before starting to implement the necessary

software architecture. While there are multiple distributed stream processing systems (DSPS) and several articles where we can find evaluations of their performance and scalability, for instance (*Dayarathna & Suzumura, 2013*; *Shukla, Chaturvedi & Simmhan, 2017*; *Hesse et al., 2021*); in this article we focus specifically on CEP engines and more particularly on one of the most widely renowned and efficient CEP engines, the Espertech Esper CEP engine, which use has been proposed in multiple domains in the last years, such as *Ma et al. (2019)*, *Zhu (2021)*, *Helal & Awad (2022)*. However, most of these proposals do not provide a performance or scalability assessment or present a very limited one. This research gap involves that developers of big data applications with Esper CEP do not have an understanding of how their system will perform in terms of both response time and scalability until they implement it and do real tests, which can lead to the implementation of under-resourced or over-dimensioned systems for the application domain in question, with the consequent loss of time and economic benefits. We firmly believe it is important to fill this gap because there are already many companies that are starting to use CEP for their IoT applications and smart cities or smart environments ones, such as for air quality control in smart ports (*Ortiz et al., 2022a*), water supply network management in smart cities (*Corral-Plaza et al., 2020*) and fall detection for smart health (*Blunda et al., 2020*). Therefore, in this work we expect to fill such research gap by carrying out a performance and scalability analysis of Espertech Esper CEP engine, integrating it with two reputable message brokers and in a centralized and distributed configuration. Our main aim is to provide developers with an analysis of various Esper implementation options, where different options are considered in terms of resource requirements, efficiency and scalability, allowing them to make a design decision before starting with the implementation of the system. It is not our aim to compare Esper to other CEP systems, but we discuss some other articles focused on such comparison in the related work section.

With this aim in mind, the following research questions have been defined:

RQ1. In a centralized architecture, with a single Esper CEP engine, what are the advantages and disadvantages of integrating it with two competing brokers such as RabbitMQ and Kafka and which one should be used to achieve the best performance in real-time stream data processing?

RQ2. When does it outweigh using a distributed CEP architecture to achieve greater horizontal scalability and how does this impact system performance?

RQ3. Which of these Esper engines and which messaging broker should I use depending on my system requirements?

To answer these questions, we have defined a series of event patterns with different operators that constitute an extension of the benchmark we presented in *Ortiz et al. (2022b)* and compared the performance and scalability of various architectures, using, as mentioned, different Espertech software products, on which the event patterns of the defined benchmark were deployed, with two widely used message brokers—RabbitMQ and Kafka, which will receive data with several input rates. Particularly, to answer RQ1 we have compared two centralized implementations, on the one hand, an open-source CEP engine integrated with RabbitMQ, and, on the other, an Esper enterprise edition with EQC and HA CEP engine with Kafka, since the use of Kafka with such engine provides additional

reliability that cannot be provided with open-source Esper and RabbitMQ. To answer RQ2 it was also necessary to integrate the latest with two distributed instances of Esper enterprise edition with EQC and HA. The comparison and analysis of the results of these three configurations led us to the discussion of RQ3. To summarize, the main contribution of this article is a performance comparison of several CEP products from Espertech, a company which offers one of the most reliable and efficient CEP engines, and a comprehensive grammar for event pattern definition, together with the well-known messaging brokers RabbiMQ and Apache Kafka. For this purpose, we provide:

- A benchmark with a variety of operators commonly used in CEP event patterns.
- The performance and resource consumption evaluation of a software architecture integrating an instance of Esper open-source CEP engine with RabbitMQ using the provided benchmark and an incremental rate of incoming events to the system.
- The performance and resource consumption evaluation of a software architecture integrating an instance of Esper enterprise edition with EQC and HA with Kafka using the same benchmark and incremental rate of incoming events.
- The performance and resource consumption evaluation of a software architecture integrating two distributed instances of Esper enterprise edition with EQC and HA with Kafka using the same benchmark and incremental rate of incoming events.
- Finally, a discussion on the results of the empirical evaluations is provided, along with the answer to the research questions previously stated. The answer to RQ3 includes a series of suggestions to consider when choosing which Espertech CEP product and which broker may be most suitable, according to the requirements in terms of resource consumption, response time, and the expected or desired rate of incoming data that the system should be able to handle, as well as reliability and fault tolerance requirements.

The rest of the article is structured as follows: First the background on the technologies used in the implementation of the architectures evaluated in the article is presented. Then the implementation of the software architectures to be evaluated are presented. Afterwards, materials and methods used for the evaluation, *i.e.*, the hardware resources used, the method followed and the benchmark proposed are explained; and then the tests results are analyzed. Following, related work is examined. Finally, the results of the evaluation are discussed and the responses to the research questions provided, to conclude with the outlined conclusions.

## BACKGROUND

In this section, we present the main technologies used in this article, namely CEP and message brokers, as well as their integration.

### Complex event processing

CEP (*Luckham, 2012*) is a robust technology by means of which we can capture, analyze and correlate huge amounts of data in real time, from different application domains and in different formats, to detect key situations in one or several specific domains at the moment

the situation occurs (*Inzinger et al., 2014*). When working with this technology, the incoming data to be processed by the system are called simple events, while the detected situations are called complex events. In order to detect these complex events, it is necessary to first define the combination of simple events that will allow the complex event to be detected. This is done by defining what is called an event pattern; this event pattern analyzes and correlates one or several simple events in a given period of time. Thus, for a particular application domain, a set of event patterns is defined to specify the conditions that must be met from the event content of one or more incoming data streams in order to detect the situations of interest within the scope of that domain. These event patterns must be deployed in a CEP engine, that is the software responsible for capturing the simple input events, analyzing in real time whether some of the event patterns deployed on the simple input event stream are fulfilled, and creating the complex output events. Typically, event patterns are defined manually, either by software programmers directly in the programming language of the CEP engine in question, or by domain experts using specific graphical tools for the automatic generation of event patterns in that language (*Corral-Plaza et al., 2021*). This is because domain experts usually have the knowledge of the situations of interest they seek to detect and the series of events that cause them; however, it may also be the case that some of these patterns are unknown, in which case CEP can be combined with machine learning techniques to try to automatically learn new CEP rules for the domain in question.

In this article, we have carried out a manual definition of patterns without combining them with machine learning techniques, since what we are interested in finding out is the cost of processing in the CEP engine and not the cost of learning new patterns in those domains in which these are unknown *a priori*. Concerning the CEP engine, we have adopted the Java-based Esper CEP engine, not only because of its high performance, versatility of operators for the event patterns and technological maturity, but also because it has not only one version for centralized processing, but, in addition, also two versions for distributed and high availability processing.

Esper (*EsperTech Inc, 2023a*) is an open-source software available under the GNU general public license (GPL) v2. This open-source version can be used as a centralized CEP engine. In addition to this open-source version, Espertech provides two closed-source options. The first one is Esper high availability (Esper HA) (*EsperTech Inc, 2023b*), which has two main capabilities: (1) it allows to save and recover the runtime state, providing high-availability in horizontal architectures. This enables several CEP engines to be deployed to solve problems such as service outages, with a possibility to recover the state of the failed engine in another of the available engines. (2) In addition, Esper HA supports memory management, with it being unnecessary to store all states in heap memory, unlike Esper open-source, which keeps all runtime in memory only. Esper HA thus has the added advantage of fast recovery from failure.

The second one is Esper enterprise edition (*EsperTech Inc, 2023c*); this provides a technology called Esper query container (EQC), which encompasses the horizontal scale-out architecture for Esper and Esper HA. EQC allows more engines to be added and removed, distributing the load dynamically, and providing greater fault tolerance—if an

engine shuts down automatically, it immediately corrects the error by sending the load to another engine. However, Esper EQC has one limitation: it must obligatorily be used with Kafka and Kafka streams, as will be explained in the following sub-section.

## Message broker

The use of a message broker is essential to facilitate the reception of simple events from various sources and their forwarding to be processed by the CEP engine. Message brokers implement an asynchronous mechanism that allows for complete decoupling of source and destination messages as well as enabling messages to be stored in the broker until they can be processed by the destination element when needed; they are widely used in the IoT domain. Although there are many messaging models, the most common mechanism used in the IoT domain is publish/subscribe, where messages are published according to a set topics and users subscribe to the topics of their interest. In particular, two models are more commonly used: queues and topics. Message queues implement a load balancing algorithm so that only one consumer receives the message; thus, the message remains in the system until the consumer is connected to process it. In the case of message topics, a standard publish/subscribe mechanism is implemented, where each published message can be processed by all subscribed consumers currently connected to the topic. There are multiple message brokers (*Singh & Verma, 2022*), but we have identified RabbitMQ and Apache Kafka as among the best rated (*Lazidis, Tsakos & Petrakis, 2022*).

RabbitMQ (*VMware Inc, 2023*) is a distributed and scalable open-source message broker that acts as middleware between producers and consumers. RabbitMQ implements the protocol AMQP (advanced message queuing protocol) which is an asynchronous message delivery protocol with delivery guarantee. RabbitMQ provides several communication models including the most commonly used: publish/subscribe mechanism and topics implementation. Event producers send simple events to the publish/subscribe queue and these are stored in RabbitMQ until a consumer retrieves it; in our case, this consumer is Esper CEP. When we process the simple events and generate a complex event, the latter is sent to the desired recipient.

Apache Kafka (*Apache Software Foundation, 2023a*) is an event streaming platform that allows to publish and subscribe to event streams, store event streams, and process event streams as they occur, or retrospectively. It therefore provides real-time continuous data processing. Along with Apache Kafka, we can use Apache Kafka streams (*Apache Software Foundation, 2023b*), which is an open-source library that adds many advantages to Apache Kafka. Among them, we can highlight that Apache Kafka stream permits the automatic inclusion of several input and output in the stream, rather than having a single stream. When integrating Apache Kafka with CEP, the messages are published in the Kafka broker and stored temporarily until they are processed by the CEP engine. The complex events are then stored, and finally all the data are submitted to the desired recipient.

## Integration of complex event processing and message brokers

As mentioned, the CEP engine must receive a stream of simple streaming events, which in the IoT domain could typically come from a message broker. The integration can be more

or less expensive, depending on the particular CEP engine used, as well as depending on the message broker used, that is, the source of the data to be integrated. While, in the past, we have made use of Enterprise service buses (ESBs), which facilitate integration, routing and connection between different artefacts in our software architectures, we have also seen how the use of these ESBs can be a burden on system performance. For this reason, this article explains in "Alternative Software Architecture for the Integration of Esper CEP with a Message Broker in a Centralized and Distributed Environment" that the integration of both artefacts is straightforward. In particular, Esper CEP provides straightforward adapters to AMQP and Kafka; this is why we plan to use RabbitMQ with the AMQP 0.9.1 protocol and Apache Kafka as brokers for integration with Esper CEP engines. As will be seen in the following section, we use them to implement several different software architectures, which will be evaluated and compared to each other.

# ALTERNATIVE SOFTWARE ARCHITECTURE FOR THE INTEGRATION OF ESPER CEP WITH A MESSAGE BROKER IN A CENTRALIZED AND DISTRIBUTED ENVIRONMENT

In this section, we explain the three different implementations evaluated in this article: two centralized implementations, on the one hand, using a unique open-source CEP engine integrated with RabbitMQ, and, on the other, integrating a unique Esper enterprise edition with EQC and HA with Kafka. Finally, a distributed option with two Esper enterprise edition with EQC and HA integrated with Kafka will also be implemented.

## Implementations integrating centralized CEP and a message broker

In this section, we explain two centralized implementations for CEP with Esper 8.8.0 (*EsperTech Inc, 2022*), the latest version the engine currently offers. As just discussed, the first centralized implementation consists of the integration of open-source Esper with RabbitMQ, and the second is based on the integration of Esper enterprise edition with EQC and HA (from now on Esper Enterprise-HA-EQC) with Apache Kafka.

Figure 1 represents the data flows in the first implementation (hereon, Configuration 1), where open-source Esper CEP is integrated with RabbitMQ. The flow is as follows: (1) the external data source sends the incoming data stream to RabbitMQ, (2) Esper receives the data stream by being subscribed to a RabbitMQ queue, (3) Esper sends the detected complex events to a new queue in RabbitMQ, and (4) the external destination machine will subscribe to the new RabbitMQ queue to receive the complex events detected in order to conduct the performance analysis.

Figure 2 represents the data flows in the second implementation (hereon Configuration 2), where the Esper Enterprise-HA-EQC CEP engine is integrated with Apache Kafka. The flow is as follows: (1) the external data source sends the incoming data stream to Apache Kafka, (2) Esper receives the data stream through the subscription to an Apache Kafka topic, (3) Esper sends the complex events detected to a new topic in Apache Kafka, and (4) the external destination machine subscribes to the new Apache Kafka topic to receive the complex events detected in order to conduct the performance analysis.

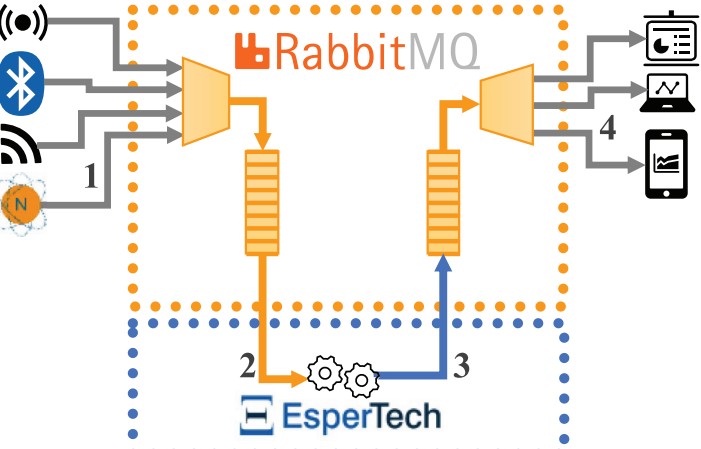

**Figure 1** Data flows for the architecture integrating open-source Esper with RabbitMQ.

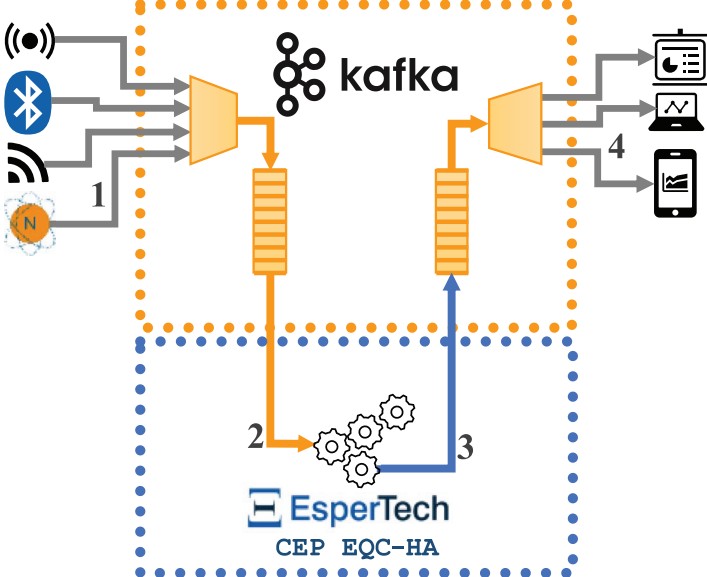

**Figure 2** Data flows for the architecture integrating one instance of Esper enterprise edition with EQC and HA with Kafka.

In both implementations, all components (incoming message broker, CEP engine and outgoing message broker) can be located on the same or different machines; in the evaluation carried out in this article, each component has been located on a different machine. As discussed, the implementations are very similar: in both cases, the data stream arrives at the messaging broker in a format readable by the CEP engine (in JSON format in our case). The CEP engine then receives the data from the broker and detects the different complex events. Finally, these complex events are sent back to the broker, from which we can extract the information we are interested in. However, it should be remembered that we used Esper open-source CEP engine in the RabbitMQ implementation, but Esper Enterprise-HA-EQC CEP engine in the Apache Kafka implementation.

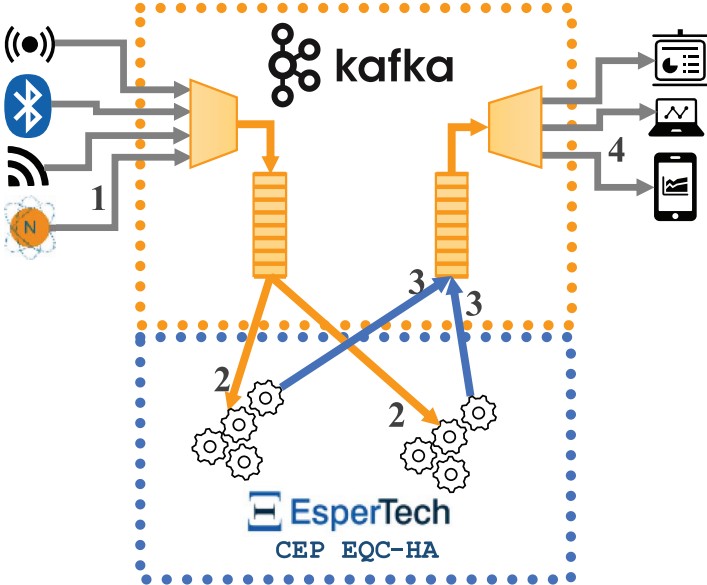

**Figure 3** Data flows for the architecture integrating two instances of Esper enterprise edition with EQC and HA with Kafka.

The use of EQC and Apache streams with Esper enterprise edition handles complex events in such a way that they natively return to the input Kafka broker. Although we could gain in memory usage by using separate brokers for the input of simple events and the output of complex events, we have adopted this architecture proposed by Esper EQC. We will follow the same criteria for the configurations of the integration of open-source Esper and RabbitMQ for the homogeneity of the tests. However, separate queues/topics are used within the broker itself. On the one hand, in RabbitMQ, we have used two queues, as there is no need to filter by topics. On the other, although Kafka streams forces the system to use topics, as there is only one incoming topic and one outgoing topic, there is no comparative decrease in performance compared to using queues in RabbitMQ for this reason.

## Implementation integrating distributed CEP and a message broker

In this section, we explain a distributed implementation for CEP with Esper 8.8.0, based on the integration of the Esper Enterprise-HA-EQC CEP engine with Apache Kafka. Figure 3 represents the data flows in the distributed implementation (hereon Configuration 3), where two instances of Esper Enterprise-HA-EQC are integrated with Apache Kafka. The flow is as follows: (1) the external data source sends the incoming data stream to Apache Kafka, (2) Esper Enterprise-HA-EQC receives the data stream by being subscribed to an Apache Kafka topic; depending of the key of each message, the message is sent to one or the other instance of the two deployed Esper Enterprise-HA-EQC CEP engines, (3) Esper Enterprise-HA-EQC CEP engine sends the detected complex events to a new topic in Apache Kafka, and (4) the external destination machine subscribes to the new Apache Kafka topic to receive the detected complex events in order to do conduct the performance analysis.

This implementation behaves in the same way as the implementations described in the previous subsection. In this case, however, we take advantage of the possibility of distributed processing by using Esper Enterprise-HA-EQC CEP engine and instantiate two CEP engines that work in a distributed way, together with Apache Kafka streams, balancing the number of events between them, and recovering the state in case one of them fails.

## MATERIALS AND METHODS

Having described the implementation of the three software architectures, we now explain the materials and methods used to evaluate their respective performance. In this section, we first describe the materials used for the tests, that is, the computer resources needed. Then we explain, on the one hand, the method followed to conduct the tests and, on the other, the benchmark event patterns defined to perform the tests.

### Computer resources

The following computer resources were used for the evaluation tests:

- A server machine with an Intel Xeon Silver 4110 processor, and 32 GB of RAM. This machine was used as broker: in the case of Configuration 1, this server hosts the RabbitMQ broker; in the case of Configurations 2 and 3, this server hosts the Kafka broker.
- A server machine with an Intel Xeon Silver 4210R processor, and 32 GB of RAM. This machine was used to host the CEP engine: it hosts the open-source Esper CEP engine in Configuration 1; it hosts Esper Enterprise-HA-EQC CEP engine in Configurations 2 and 3.
- A server machine with an Intel Xeon Silver 4210R processor, and 32 GB of RAM. This machine was used as the Esper Enterprise-HA-EQC CEP engine in the distributed implementation (configuration 3).
- A PC with an Intel i3 3220T and 4 GB of RAM. This machine was used as the external source to send data to the inbound queue.
- A PC with an Intel Core i7 8750H and 16 GB of RAM. This machine was used as the external destination machine to pull the data from the output queue and to analyze it.
- All tests were performed within the University of Cadiz network.

Note that with the proposed architectures there is no additional communication cost for the distributed implementation. This is because there is no communication between the various CEP engines, but rather it is Kafka that communicates the corresponding data to each of the CEP engines, in a similar way to how RabbitMQ or Kafka itself communicates with the single CEP engine in the centralised versions. Additional communications would only be required if a machine failed, and we wanted to recover the system. For this purpose, Esper HA, when using one or more CEP engines, makes use of the Kafka changelog file to restore the system. The changelog sets a checkpoint periodically (by default 30 s), updating

the status of the running engines up to that moment and could be used to continue processing on that or another machine from the last checkpoint.

## Methods

The tests consist of deploying a series of event patterns—described in the following subsection—in the CEP engine for each of the three configurations described above. We first evaluate the performance of the engine by deploying the event patterns separately and then deploy the whole set of event patterns at once. For each event pattern, we progressively increase the number of events sent to the CEP engine per second—for 100, 1,000, 5,000, 10,000, 15,000 and 20,000 events/s, until we reach the maximum number supported during the testing time. When reaching the limit of input events at which each configuration responds appropriately, higher event rates were not tested for such an input rate. During the test, we measure the following key performance indicators:

- CPU usage: percentage of CPU usage during the test execution in isolation from other processes.
- RAM usage: Megabytes (MB) of memory usage during the test execution in isolation from other processes.
- Latency: average time taken by the CEP engine to process each incoming event in milliseconds (ms).
- Throughput Time: total time taken by the engine to process all the incoming events in minutes (min). It allows us to quickly identify when the system becomes overloaded.

Additional time measurements are taken to check that the system is processing the events correctly, but it is considered unnecessary to detail all these times in this article.

Each of these tests is carried out for a short period of time; specifically, in our case, for 11 min. Once the short test is carried out for all the event patterns separately, the short test is performed once all the event patterns are deployed in the CEP engine. With the information on the latter, we assess the input event rate of the shortest test for which the system remains stable. We then proceed with a long test of 61 min for all the event patterns together to see if the system remains stable for a longer period of time.

We used a synthetic data emulator—nITROGEN (*Garcia-de-Prado, 2020*)—to emulate a series of data with which we ensure that a specific percentage of events meet the event patterns in the tests, therefore, the latter will be homogeneous for all rates of input events and patterns and would be reproducible if necessary. More particularly, according to the data emulated, 1 in 5 data sent to the input broker cause the event pattern conditions to be met in the case of short tests and 1 in 5,000 cause the event pattern conditions to be met in the case of long tests. Note that 1 in 5 is a very high number of complex outgoing events for the long test and we would overload the outgoing broker before being able to evaluate whether our CEP engine is able to process the event load over a long period of time, which is the key interest in this evaluation. If necessary, we could always add more brokers to process the complex outgoing events.

## Benchmark event patterns

In this section, we define the types of events and the event patterns used in the proposed tests. Event types are defined in Esper by using schemas; specifically, we defined the *BenchmarkEvent* schema shown in Listing 1. As shown below, the *BenchmarkEvent* schema contains the following fields:

- *id*: event identifier.
- *Attr1*, *attr2*, *attr3*: these are event attributes that take different values to test the different event patterns and their functionality.
- *timestamp*: this field stores the timestamp when the simple events is created.
- *esperInputTimestamp*: this field stores the timestamp when the simple event is sent to the CEP engine.

*Listing 1. Benchmark type of events*

```
Create schema BenchmarkEvent
 (id string, attr1 string, attr2 string, attr3 string, timestamp long,
esperInputTimestamp long)
```

As mentioned, the emulated data allowed the existing conditions—when a condition is set—to be met in the following event patterns in one out of every five simple events generated.

The event patterns have been chosen to test the most commonly used operators in Esper according to our experience in real domains, some of which require more memory consumption (for instance data or temporary windows) and others burden more CPU consumption (such as a GROUP BY operator).

**Statement 1**

In this event pattern, as shown in Listing 2, all messages are selected with no condition. This means a complex event will be generated for each simple event received.

*Listing 2. Statement 1 implementation*

```
@Name('Statement1')
INSERT INTO Statement1
SELECT *
FROM BenchmarkEvent
```

**Statement 2**

In this event pattern, we add a condition to statement 1 so that we increase the complexity of the event pattern. As shown in Listing 3, a complex event is generated when the incoming event has the value *Attribute* in the field attr1. As discussed, this condition will only be fulfilled in a fifth of the simple events reaching the CEP engine.

*Listing 3. Statement 2 implementation*

```
@Name('Statement2')
INSERT INTO Statement2
SELECT *
FROM BenchmarkEvent
WHERE attr1="Attribute"
```

### Statement 3

In this event pattern, we add a length sliding window of size 10 (see Listing 4) to Statement 2, so that we increase the memory usage of the event pattern. This type of window is a length sliding window that keeps the specified number of events in memory. This means that, in our case, this window keeps the last 10 events that meet the condition of having *Attribute* in the field attr1. Thus, one complex event is generated for every simple event meeting the condition in every 10-event window.

*Listing 4. Statement 3 implementation*

```
@Name('Statement3')
INSERT INTO Statement3
SELECT *
FROM BenchmarkEvent.win:length(10)
WHERE attr1="Attribute"
```

### Statement 4

In this event pattern, we add a temporal 2-min sliding window (see Listing 5) to Statement 2, so that we increase the complexity of the event pattern, but, on this occasion, using 2-min windows, which requires greater memory than 10-event windows. This type of window is a temporal sliding window that extends the time interval specified in the event pattern based on the system time. This means that, in our case, this window keeps the events that meet the conditions for 2 min from the first event that meets the condition of value atr1 equal to *Attribute*. Thus, one complex event is generated for every simple event meeting the condition in every 2-min window.

*Listing 5. Statement 4 implementation*

```
@Name('Statement4')
INSERT INTO Statement4
SELECT *
FROM BenchmarkEvent.win:time(2 min)
WHERE attr1="Attribute"
```

### Statement 5

As shown in Listing 6, in this event pattern, we add a clause COUNT DISTINCT to the time window in Statement 4, so that we increase the complexity of the event pattern, together with the memory usage. Thus, a complex event is generated for all the simple events produced in the last 2 min that have a field attr3 with value *Attribute*, which will give us the number of events with a different attr2 for every temporal window of 2 min.

*Listing 6. Statement 5 implementation*

```
@Name('Statement5')
INSERT INTO Statement5
SELECT COUNT (DISTINCT attr2) as eventTotal
FROM BenchmarkEvent.win:time(2 min)
WHERE attr3="Attribute"
```

### Statement 6

As shown in Listing 7, in this event pattern, we add a GROUP BY clause to Statement 5 so that we further increase the complexity of the event pattern. This allows us to separate

the events depending on the attr1 value. Thus, a complex event is generated for all the event produced in the last 2 min that have a field attr3 with value *Attribute*, grouped by attr1.

*Listing 7. Statement 6 implementation*

```
@Name('Statement6')
INSERT INTO Statement6
SELECT COUNT (DISTINCT attr2) as eventTotal
FROM BenchmarkEvent.win:time(2 min)
WHERE attr3="Attribute"
GROUP BY attr1
```

**Statement 7**

In this event pattern, we replace the time sliding window in Statement 5 with a size-10 length sliding window (see Listing 8), so that we increase the complexity of the event pattern, together with the memory usage. Thus, a complex event is generated for every simple event that accomplishes the condition of field attr3 with value *Attribute* in every 10-event window.

*Listing 8. Statement 7 implementation*

```
@Name('Statement7')
INSERT INTO Statement5
SELECT COUNT (DISTINCT attr2) as eventTotal
FROM BenchmarkEvent.win:length(10)
WHERE attr3="Attribute"
```

**Statement 8**

To finish, we replace the time sliding window in Statement 6 with a size-10 length sliding window (see Listing 9), so that we further increase the complexity of the event pattern. Thus, a complex event is generated for every simple event that accomplishes the condition of field attr3 with value *Attribute* grouped by attr1 in every 10-event window.

*Listing 9. Statement 8 implementation*

```
@Name('Statement8')
INSERT INTO Statement8
SELECT COUNT (DISTINCT attr2) as eventTotal
FROM BenchmarkEvent.win:length(10)
WHERE attr3="Attribute"
GROUP BY attr1
```

## RESULTS

This section first explains the preprocessing of the raw data obtained in the tests for further assessment, and then explains the results and their assessment.

### Preprocessing the data obtained from the tests

To be able to evaluate the results and show them in a comprehensible way in the body of the article, a preprocessing of the raw data obtained during the tests has been carried out. The result of this preprocessing, which has served as the basis for the subsequent results

assessment section, is provided as Supplemental Material. In this Supplemental Material a number of spreadsheets can be found, in particular one spreadsheet for each set of tests, *i.e.*, one spreadsheet for each short test of each benchmark event pattern, one with the short test of all event patterns and one with the long test of all event patterns. In each of these spreadsheets we can find a series of tabs representing each of the configurations (Configuration 1, Configuration 2 and Configuration 3) with each of the input rates (100, 1,000, 5,000, 10,000, 15,000 and 20,000 events per second or up to the maximum input rate supported in each case). Each of these tabs includes measurements and calculations for each parameter analyzed (latency, percentage of CPU usage and memory usage, as described in the Methods section).

These spreadsheets have been populated after preprocessing the huge set of raw data obtained in each test; note that, for example, in a short test of 11 min with an input rate of 100 events per second we could obtain up to 66,000 latency data and in the same short test with an input rate of 20,000 events per second we could obtain up to 13,200,000 latency data. This is why the average of the data received in each second has been calculated during the preprocessing, which is the average value that has been included for latency, CPU usage and memory usage in the spreadsheets. Depending on the input rate tested and the corresponding complex events detected for each of the evaluated event pattern, more or less latency data per second have been used to calculate the average; the spreadsheet indicates in its *Cont* column the number of data used for the average of each second for each configuration, input rate and event pattern; in addition, the maximum and minimum value of the data set of that second is also indicated, as well as the standard deviation of each second and the average ± the standard deviation. In the short tests, for the CPU and memory measurements, only one sample is taken per second so as not to slow down the system with such a data collection, so the average coincides with the only sample of that second and the standard deviation is zero, since, as we have said, there is only one sample. In the long test the same has been done, but grouping the data every 10 s, so the average latency the average for all the data measured during every 10 s is calculated, and the average of CPU and memory of the 10 available data every second is calculated; therefore, the standard deviation is no longer zero.

In the following subsection, these average data will be the one represented in the figures. Besides, to make it easier to understand the data at a glance and to compare the data for each configuration, average latency, CPU and memory consumption of the data obtained for all seconds of each test duration has been included in a set of tables in the following subsection.

## Results assessment

Based on the preprocessing of the raw data obtained during the tests, this section presents the results obtained in the performance tests with Configurations 1, 2 and 3 with each of the event patterns explained in the benchmark event patterns section separately and together for a short period of time (11 min) and for a longer one (61 min) for all the event patterns together, as explained in the Methods section. We sought to evaluate performance from an incoming event rate of 100 events/s, progressively, as previously explained, up to

**Table 1 Performance of statement 1 for configurations 1, 2 and 3.**

| Incoming rate (Events/s) | Configuration 1 | | | | Configuration 2 | | | | Configuration 3 | | | |
|---|---|---|---|---|---|---|---|---|---|---|---|---|
| | Throughput time (min) | Memory usage (MB) | CPU usage (%) | Latency (ms) | Throughput time (min) | Memory usage (MB) | CPU usage (%) | Latency (ms) | Throughput time (min) | Memory usage (MB) | CPU usage (%) | Latency (ms) |
| 100 | 11 | 468 | 0.23 | 0.07 | 11 | 521.5 | 0.6 | 0.077 | 11 | 518.9 523.8 | 0.39 0.4 | 0.072 |
| 1,000 | 11 | 475.5 | 1.73 | 0.09 | 11 | 551.1 | 2.2 | 0.13 | 11 | 547.7 544.6 | 1.46 1.46 | 0.093 |
| 5,000 | 11 | 482.4 | 3.45 | 0.046 | 11 | 594.6 | 2.34 | 0.23 | 11 | 608.5 594.2 | 2.95 2.87 | 0.099 |
| 10,000 | 11:11 | 532 | 3.32 | 0.017 | 22 | 606 | 2.12 | 0.19 | 11 | 580.1 586.6 | 2.88 2.89 | 0.124 |
| 15,000 | – | – | – | – | – | – | – | – | 42 | 604.5 664.6 | 0.87 0.87 | 0.46 |
| 20,000 | – | – | – | – | – | – | – | – | – | – | – | – |

20,000 events/s. When reaching the limit of input events at which each configuration responds appropriately, higher event rates were not tested for this input rate, and those cells in the tables thus appear with values (−). It should be noted that at 15,000 events per second and above, the message brokers, together with the capacity of the network, cause a delay in execution and may be responsible for the instability of the system. However, the CEP engine could handle a greater number of events/s in isolation, but we must evaluate the architecture as a whole.

As explained, the event pattern in Listing 2 generates a complex event for each simple event entering the system. Table 1 shows the results of the tests performed with Statement 1 for the three configurations. After performing the tests, we were able to draw the conclusion that, in the centralized architectures (Configurations 1 and 2), the integration of RabbitMQ with open-source Esper performs better than integrating Esper Enterprise-HA-EQC with Kafka. As shown in the table, the latter cannot properly deal with an input rate of 10,000 events per second since it takes 22 min to finish processing an 11-min data entrance, *vs* 11 min and 11 s of the open-source version. We can also observe a poorer average latency for Esper Enterprise-HA-EQC version compared to the open-source one. The reason for the lower performance may be the backups that Esper HA makes to be able to recover the system after a failure. However, in both implementations, the tests show similar memory and CPU consumption, although always slightly better for the open-source version. In the case of the distributed architecture (Configuration 3), although the performance is not better, with an average latency of 0.124 ms for an input of 10,000 events/s compared to 0.017 ms for open-source Esper, it can complete the execution in the 11 min that the test should take. In this case, CPU usage is slightly better in the distributed version than in the open source, although memory seems to be better managed by the latter. We tested whether the distributed configuration could cope with an input of 15,000 events/s, but, in this case, the execution took 42 min to finish. In the case of Configuration 2 it was inappropriate to increase the incoming event rate as it was not able to handle the rate of 10,000 events/s; and in the case of Configuration 1, although 11 min and 11 s is a reasonable throughput time, it already shows that we cannot increase the incoming data rate.

**Table 2 Performance of statement 2 for configurations 1, 2 and 3.**

| Incoming rate (Events/s) | Configuration 1 | | | | Configuration 2 | | | | Configuration 3 | | | |
|---|---|---|---|---|---|---|---|---|---|---|---|---|
| | Throughput time (min) | Memory usage (MB) | CPU usage (%) | Latency (ms) | Throughput time (min) | Memory usage (MB) | CPU Usage (%) | Latency (ms) | Throughput time (min) | Memory usage (MB) | CPU usage (%) | Latency (ms) |
| 100 | 11 | 483.9 | 0.16 | 0.052 | 11 | 523.6 | 0.54 | 0.088 | 11 | 638.3 560.7 | 0.37 0.35 | 0.085 |
| 1,000 | 11 | 486.6 | 1.34 | 0.0178 | 11 | 539.3 | 1.76 | 0.09 | 11 | 530.7 554.4 | 1.2 1.22 | 0.077 |
| 5,000 | 11 | 496.9 | 2.64 | 0.0615 | 11 | 596.6 | 2.97 | 0.089 | 11 | 559.5 562.3 | 2.59 2.58 | 0.061 |
| 10,000 | 11 | 545.2 | 3.78 | 0.045 | 11 | 691.79 | 3.41 | 0.04 | 11 | 554.3 596.6 | 2.97 2.91 | 0.046 |
| 15,000 | 11 | 558.3 | 3.76 | 0.021 | 14 | 671.6 | 2.32 | 0.094 | 11 | 579.5 604.1 | 3.27 3.18 | 0.027 |
| 20,000 | 13 | 547.6 | 2.14 | 0.0104 | – | – | – | – | 11:15 | 645.6 591.3 | 2.41 2.23 | 0.092 |

**Table 3 Performance of statement 3 for configurations 1, 2 and 3.**

| Incoming rate (Events/s) | Configuration 1 | | | | Configuration 2 | | | | Configuration 3 | | | |
|---|---|---|---|---|---|---|---|---|---|---|---|---|
| | Throughput time (min) | Memory usage (MB) | CPU usage (%) | Latency (ms) | Throughput time (min) | Memory usage (MB) | CPU usage (%) | Latency (ms) | Throughput time (min) | Memory usage (MB) | CPU usage (%) | Latency (ms) |
| 100 | 11 | 431 | 0.2 | 0.057 | 11 | 586.1 | 0.53 | 0.08 | 11 | 591.1 672.8 | 0.35 0.36 | 0.086 |
| 1,000 | 11 | 495 | 1.37 | 0.014 | 11 | 543.8 | 1.79 | 0.092 | 11 | 585.3 541.9 | 1.27 1.26 | 0.101 |
| 5,000 | 11 | 507.9 | 2.66 | 0.041 | 11 | 630.6 | 2.97 | 0.087 | 11 | 625.8 617.7 | 2.58 2.62 | 0.091 |
| 10,000 | 11 | 518.3 | 3.68 | 0.002 | 11 | 606.3 | 3.47 | 0.053 | 11 | 566.5 585.6 | 2.92 2.94 | 0.031 |
| 15,000 | 11 | 518.6 | 4.72 | 0.029 | 15 | 609.3 | 2.16 | 0.085 | 11 | 621.3 647 | 2.96 2.97 | 0.052 |
| 20,000 | 13 | 546.4 | 2.25 | 0.01 | – | – | – | – | 12 | 626.5 617.8 | 2 1.97 | 0.092 |

As explained, the event pattern in Listing 3 (Statement 2) generates a complex event for one out of every five incoming events. Table 2 shows the results of the tests performed with Statement 2 for the three configurations. For the three configurations, we can see a similar behavior in question of memory and CPU consumption. In this case, by having less complex generated events than in the previous statement, both Configurations 1 and 3 were able to perfectly handle an input rate of 15,000 events/s, with an average latency of 0.021 ms per event for the open source Esper *vs* 0.027 ms for the distributed Esper Enterprise-HA-EQC implementation. However, Configuration 2 was unable to handle it correctly, taking 14 min to finish the test. Neither Configuration 1 nor Configuration 3 were able to properly finish the execution with an input event rate of 20,000 events/s.

The event pattern in Listing 4 (Statement 3) generates a complex event for every simple one that accomplishes the condition in every 10-event window, with only 1 out of 5 incoming events meeting the condition. Table 3 presents the results of the tests performed with Statement 3 for the three configurations. Again, for all the configurations, we can see a similar behavior in terms of memory, but a higher, albeit not significant, CPU consumption for Configuration 1. Note that we do not consider CPU consumption of up to 10% to be significant, although using less powerful machines may have a greater impact on CPU consumption. In this case, both Configurations 1 and 3 were perfectly able to handle an input rate of 15,000 events/s, with an average latency of 0.029 ms for the open-source Esper *vs* 0.052 ms for the distributed Esper Enterprise-HA-EQC implementation.

**Table 4 Performance of statement 4 for configurations 1, 2 and 3.**

| Incoming rate (Events/s) | Configuration 1 | | | | Configuration 2 | | | | Configuration 3 | | | | | |
|---|---|---|---|---|---|---|---|---|---|---|---|---|---|---|
| | Throughput time (min) | Memory usage (MB) | CPU usage (%) | Latency (ms) | Throughput time (min) | Memory usage (MB) | CPU usage (%) | Latency (ms) | Throughput time (min) | Memory usage (MB) | | CPU usage (%) | | Latency (ms) |
| 100 | 11 | 487.9 | 0.2 | 0.005 | 11 | 675.9 | 0.58 | 0.093 | 11 | 539.1 | 518 | 0.38 | 0.37 | 0.104 |
| 1,000 | 11 | 661.2 | 1.43 | 0.117 | 11 | 711.6 | 1.95 | 0.112 | 11 | 566.6 | 573.7 | 1.32 | 1.31 | 0.082 |
| 5,000 | 11 | 693.2 | 3 | 0.0015 | 11 | 1,011.1 | 4.04 | 0.089 | 11 | 791.2 | 782.3 | 3 | 3.03 | 0.087 |
| 10,000 | 11 | 760.5 | 5.17 | 0.032 | 11 | 1,536 | 5.31 | 0.078 | 11 | 949.2 | 1,024 | 3.96 | 3.97 | 0.068 |
| 15,000 | 11 | 1,004.6 | 6.35 | 0.0114 | 16 | 3,072 | 3.61 | 0.083 | 11 | 1,740.8 | 1,945.6 | 4.54 | 4.56 | 0.041 |
| 20,000 | 14 | 997.9 | 8.1 | 0.0106 | – | – | – | – | 11:27 | 1,945.6 | 1,945.6 | 3.68 | 3.59 | 0.086 |

**Table 5 Performance of statement 5 for configurations 1, 2 and 3.**

| Incoming rate (Events/s) | Configuration 1 | | | | Configuration 2 | | | | Configuration 3 | | | | | |
|---|---|---|---|---|---|---|---|---|---|---|---|---|---|---|
| | Throughput time (min) | Memory usage (MB) | CPU usage (%) | Latency (ms) | Throughput time (min) | Memory usage (MB) | CPU usage (%) | Latency (ms) | Throughput time (min) | Memory usage (MB) | | CPU usage (%) | | Latency (ms) |
| 100 | 11 | 493.5 | 0.18 | 0.0079 | 11 | 594 | 0.58 | 0.102 | 11 | 522.3 | 522.1 | 0.36 | 0.38 | 0.106 |
| 1,000 | 11 | 677.7 | 1.43 | 0.0315 | 11 | 729.5 | 1.96 | 0.11 | 11 | 591 | 653.5 | 1.35 | 1.33 | 0.097 |
| 5,000 | 11 | 723.6 | 3.02 | 0.0034 | 11 | 1,008.4 | 4.01 | 0.06 | 11 | 779.6 | 770.5 | 3 | 3.01 | 0.091 |
| 10,000 | 11 | 727.9 | 4.99 | 0.0179 | 11 | 1,536 | 5.19 | 0.043 | 11 | 1,126.4 | 1,024 | 3.9 | 3.9 | 0.059 |
| 15,000 | 11 | 968 | 9.78 | 0.014 | 15 | 2,560 | 3.62 | 0.08 | 11 | 1,433.6 | 1,331.1 | 4.48 | 4.44 | 0.029 |
| 20,000 | 15 | 1,126.4 | 6.72 | 0.0109 | – | – | – | – | 11:18 | 2,764.8 | 1,740.8 | 3.75 | 3.14 | 0.11 |

However, Configuration 2 was unable to handle it correctly, taking 15 min to finish the test.

The event pattern in Listing 5 (Statement 4) generates a complex event for all events that meet the condition during 2 min, with only 1 out of 5 incoming events meeting the condition. Table 4 shows the results of the tests performed with Statement 4 for the three configurations. As in the previous case, we can see a higher, albeit not significant, CPU consumption for Configuration 1, but, in this case, we see that Configuration 1 is considerably more efficient in terms of memory usage than the other 2 configurations: 1,004.6 MB used in Configuration 1 compared to 3,072 MB in Configuration 2 and an average of 1,843 MB from the two distributed machines in Configuration 3 (we have discarded decimals for both CPU and memory usage in the explanation as they are not significant) with an input rate of 15,000 events/s. Although we have more than sufficient RAM memory on the servers on which we deployed Esper, we do feel it important to highlight the increase in memory for some of the configurations. Both Configurations 1 and 3 were able to perfectly deal with an input rate of 15,000 events/s, with an average latency of 0.0114 ms for the open-source Esper *vs* 0.041 ms for the distributed Esper Enterprise-HA-EQC implementation. However, Configuration 2 was unable to handle it properly, taking 16 min to finish the test.

**Table 6 Performance of statement 6 for configurations 1, 2 and 3.**

| Incoming rate (Events/s) | Configuration 1 | | | | Configuration 2 | | | | Configuration 3 | | | | |
|---|---|---|---|---|---|---|---|---|---|---|---|---|---|
| | Throughput time (min) | Memory usage (MB) | CPU usage (%) | Latency (ms) | Throughput time (min) | Memory usage (MB) | CPU usage (%) | Latency (ms) | Throughput time (min) | Memory usage (MB) | | CPU usage (%) | | Latency (ms) |
| 100 | 11 | 423.1 | 0.2 | 0.035 | 11 | 709.1 | 0.58 | 0.098 | 11 | 529.8 | 509.7 | 0.38 | 0.38 | 0.098 |
| 1,000 | 11 | 696.6 | 1.38 | 0.0026 | 11 | 736 | 2.02 | 0.12 | 11 | 578.6 | 597.7 | 1.31 | 1.31 | 0.103 |
| 5,000 | 11 | 712.4 | 2.99 | 0.075 | 11 | 1,024 | 4.06 | 0.076 | 11 | 877.5 | 865.5 | 3.06 | 3.04 | 0.097 |
| 10,000 | 11 | 746.9 | 4.9 | 0.038 | 11 | 2,252.8 | 4.89 | 0.037 | 11 | 1,018.2 | 1,126.4 | 3.94 | 3.78 | 0.066 |
| 15,000 | 11 | 924.2 | 9.87 | 0.041 | 14 | 2,764.8 | 3.99 | 0.086 | 11 | 1,331.2 | 1,331.2 | 4.65 | 4.55 | 0.041 |
| 20,000 | 15 | 1,001.1 | 6.55 | 0.0447 | – | – | – | – | 11:04 | 2,252.8 | 1,945.6 | 4.28 | 3.31 | 0.081 |

The event pattern in Listing 6 (Statement 5) generates a complex event for every 2-min temporal window that has field attr3 with value *Attribute*, but we only store the value of the account, not all attributes of the event. Table 5 shows the results of the tests performed with Statement 5 for the three configurations. In this test, we can see a similar behavior in terms of CPU consumption as in the two previous statements; and, as in the case of Statement 4, we see that Configuration 1 is significantly more efficient in terms of memory usage than the other two configurations: 968 MB used in Configuration 1 *vs* 2,560 MB and an average of 1,382 MB in Configurations 2 and 3, respectively, for an input rate of 15,000 events/s. Although this event pattern generates a smaller number of events than the previous one, the reason for the high memory consumption is that 2-min windows with all the events meeting the condition have to be kept in memory during the whole test run. Configurations 1 and 3 were both able to perfectly handle an input of 15,000 events/s, with an average of 0.014 ms of latency for the open-source Esper *vs* 0.029 ms for the distributed Esper Enterprise-HA-EQC implementation. However, configuration 2 was unable to handle it correctly, taking 15 min to finish the test.

The event pattern in Listing 7 (Statement 6) generates a complex event for every 2-min temporal window grouped by a particular attribute, but we only store the value of the account, not all attributes of the event. In Table 6, we can find the results of the tests performed with Statement 6 for the three configurations. Again, we can find the same behavior as in previous statements: a higher but not significant CPU consumption for Configuration 1 and considerably better efficiency in terms of memory usage for Configuration 1 than for the other 2 configurations: 942.2 MB used in Configuration 1 *vs* 2,764.8 MB and 1,331.2 MB in Configurations 2 and 3, respectively, for an input of 15,000 events/s. Configurations 1 and 3 were both able to perfectly handle an input rate of 15,000 events/s, with an average of 0.041 ms of latency in both implementations. However, Configuration 2 was unable to handle it properly, taking 15 min to finish the test.

The event pattern in Listing 8 (Statement 7) generates a complex event for every 10 simple events that accomplish the condition, but we only store the value of the account, not all attributes of the event. Table 7 details the results of the tests performed with Statement 7 for the three configurations. In this case, we can see similar CPU consumption and

**Table 7 Performance of statement 7 for configurations 1, 2 and 3.**

| | Configuration 1 | | | | Configuration 2 | | | | Configuration 3 | | | |
|---|---|---|---|---|---|---|---|---|---|---|---|---|
| Incoming rate (Events/s) | Throughput time (min) | Memory usage (MB) | CPU usage (%) | Latency (ms) | Throughput time (min) | Memory usage (MB) | CPU usage (%) | Latency (ms) | Throughput time (min) | Memory usage (MB) | CPU usage (%) | Latency (ms) |
| 100 | 11 | 383.3 | 0.18 | 0.003 | 11 | 599.3 | 0,55 | 0.093 | 11 | 684.3  577.8 | 0.36  0.36 | 0.097 |
| 1,000 | 11 | 508.7 | 1.36 | 0.011 | 11 | 555.2 | 1.78 | 0.097 | 11 | 533.6  544.4 | 1.2  1.28 | 0.091 |
| 5,000 | 11 | 519.1 | 2.67 | 0.056 | 11 | 594.6 | 2.92 | 0.06 | 11 | 544.7  572.4 | 2.59  2.58 | 0.084 |
| 10,000 | 11 | 528 | 3.86 | 0.0021 | 11 | 650.4 | 3.38 | 0.059 | 11 | 598  544.1 | 2.98  2.95 | 0.05 |
| 15,000 | 11 | 532.6 | 2.92 | 0.037 | 16 | 667.9 | 2.19 | 0.11 | 11 | 641.4  587.9 | 3.25  3.24 | 0.037 |
| 20,000 | 14 | 563.1 | 2.05 | 0.033 | – | – | | – | 11:05 | 615.2  609.9 | 2.74  2.75 | 0.084 |

**Table 8 Performance of statement 8 for configurations 1, 2 and 3.**

| | Configuration 1 | | | | Configuration 2 | | | | Configuration 3 | | | |
|---|---|---|---|---|---|---|---|---|---|---|---|---|
| Incoming rate (Events/s) | Throughput time (min) | Memory usage (MB) | CPU usage (%) | Latency (ms) | Throughput time (min) | Memory usage (MB) | CPU usage (%) | Latency (ms) | Throughput time (min) | Memory usage (MB) | CPU usage (%) | Latency (ms) |
| 100 | 11 | 390.5 | 0.19 | 0.024 | 11 | 648.3 | 0.51 | 0.095 | 11 | 520.7  520.3 | 0.36  0.37 | 0.107 |
| 1,000 | 11 | 506.5 | 1.36 | 0.013 | 11 | 540.7 | 1.82 | 0.102 | 11 | 566.2  549.3 | 1.23  1.27 | 0.097 |
| 5,000 | 11 | 499.7 | 2.71 | 0.004 | 11 | 577.2 | 2.95 | 0.049 | 11 | 599.7  655.8 | 2.58  2.56 | 0.096 |
| 10,000 | 11 | 518.8 | 3.76 | 0.033 | 11 | 609.1 | 3.49 | 0.053 | 11 | 605.7  592.5 | 2.92  2.90 | 0.076 |
| 15,000 | 11 | 540.3 | 4.99 | 0.031 | 15 | 759.7 | 2.27 | 0.102 | 11 | 658.5  608.7 | 3.23  3.21 | 0.036 |
| 20,000 | 14 | 546.4 | 2.1 | 0.0298 | – | – | – | – | 11:01 | 639.5  611.5 | 2.31  2.34 | 0.101 |

memory usage for the three configurations. Configurations 1 and 3 were both able to perfectly handle an input rate of 15,000 events/s, with an average latency of 0.037 ms in both implementations. However, Configuration 2 was unable to handle it, taking 16 min to finish the test.

The event pattern in Listing 9 (Statement 8) generates a complex event for every 10 simple events that accomplish the condition grouped by a particular attribute, but we only store the value of the account, not all attributes of the event. Table 8 presents the results of the tests performed with Statement 8 for the three configurations. As in the previous statement, we have similar memory usage for all three configurations, and slightly higher but not significant CPU usage in Configuration 1. In addition, Configurations 1 and 3 were both able to handle an input rate of 15,000 events/s, with an average latency of 0.031 and 0.036 ms, respectively. However, Configuration 2 was unable to handle it correctly, taking 16 min to finish the test.

As explained in the Methods section, we also performed the test by deploying all event patterns at the same time. Table 9 shows similar CPU consumption for all configurations, although the open-source Esper version (configuration 1) shows a significant increase in CPU consumption during execution with an input rate of 15,000 events/s. However, it is Configuration 1 that still maintains the best performance in terms of memory usage:

**Table 9 Performance of all statements for configurations 1, 2 and 3.**

| | Configuration 1 | | | | Configuration 2 | | | | Configuration 3 | | | | | |
|---|---|---|---|---|---|---|---|---|---|---|---|---|---|---|
| Incoming rate (Events/s) | Throughput time (min) | Memory usage (MB) | CPU usage (%) | Latency (ms) | Throughput time (min) | Memory usage (MB) | CPU usage (%) | Latency (ms) | Throughput time (min) | Memory usage (MB) | | CPU usage (%) | | Latency (ms) |
| 100 | 11 | 541.6 | 0.25 | 0.011 | 11 | 669.7 | 0.63 | 0.2 | 11 | 553.5 | 569.6 | 0.4 | 0.41 | 0.21 |
| 1,000 | 11 | 744.1 | 1.9 | 0.024 | 11 | 780.2 | 2.44 | 0.21 | 11 | 704.8 | 718.7 | 1.46 | 1.56 | 0.2 |
| 5,000 | 11 | 765.3 | 4.41 | 0.003 | 11 | 1,740.8 | 4.71 | 0.12 | 11 | 1,228.8 | 1,228.8 | 3.73 | 3.7 | 0.14 |
| 10,000 | 11 | 879 | 6.87 | 0.089 | 11 | 3,379.2 | 6.38 | 0.103 | 11 | 1,433.6 | 1,433.6 | 4.4 | 4.42 | 0.074 |
| 15,000 | 11 | 1,228.8 | 13.85 | 0.049 | 11:05 | 3,276.8 | 7.64 | 0.11 | 11 | 3,072 | 3,072 | 5 | 5.15 | 0.062 |
| 20,000 | 12 | 1,331.2 | 11.5 | 0.044 | | | | | 11 | 2,355.2 | 2,252.8 | 5.65 | 5.57 | 0.13 |

**Table 10 Performance on long test of all statements together for configuration 1, 2 and 3.**

| | Configuration 1 | | | | Configuration 2 | | | | Configuration 3 | | | | | |
|---|---|---|---|---|---|---|---|---|---|---|---|---|---|---|
| Incoming rate (Events/s) | Throughput time (min) | Memory usage (MB) | CPU usage (%) | Latency (ms) | Throughput time (min) | Memory usage (MB) | CPU usage (%) | Latency (ms) | Throughput time (min) | Memory usage (MB) | | CPU usage (%) | | Latency (ms) |
| 10,000 | Test not performed | | | | 61 | 3,072 | 5.12 | 0.075 | Test not performed | | | | | |
| 15,000 | 61 | 1,331.2 | 6.56 | 0.019 | Unsuccessful test | | | | Test not performed | | | | | |
| 20,000 | Unsuccessful test | | | | Test not performed | | | | 61 | 2,252.8 | 1,945.6 | 5.22 | 5.39 | 0.047 |

1,228.8 MB with the open-source Esper version *vs* 3,072 MB on the distributed machines. Configuration 2 did not successfully terminate the execution for an input rate of 15,000 events per second, but Configurations 1 and 3 did.

To see whether the system remains stable over time, we performed a longer test with all the event patterns deployed. In this case, as shown in Table 10, Configuration 1 was unable to cope with the constant load of 20,000 events/s. Only Configuration 3 was able to cope, while Configuration 1 managed to reach 15,000 events/s and Configuration 2 only 10,000 events/s. Configuration 1 was able to handle 15,000 events/s with 6.56% of CPU usage, 1,331 MB of memory usage and an average processing time per event of 0.019 ms. Configuration 2 was able to handle 10,000 events/s also with a low CPU usage of 5.12%, but with a high memory usage of 3,072 MB and an average latency of 0.075 ms. Finally, the execution with the input rate of 20,000 events per second of Configuration 3 also remained at a low CPU consumption (5%) but with a somewhat high memory consumption (2,103.7 MB average) and an adequate latency of 0.047 ms.

To visualize the results more clearly, Figs. 4–6 have been added. Since representing data for each second did not permit to visualized data well due to their density, such figures represent the average data for every 10 s of execution. In particular, the average processing time of each event for every 10 s of execution is shown in Fig. 4. It clearly shows how the open-source Esper option with RabbitMQ presents better performance, although it is not able to reach as high rates of incoming events per second as the option with distributed Esper Enterprise-HA-EQC with two machines.

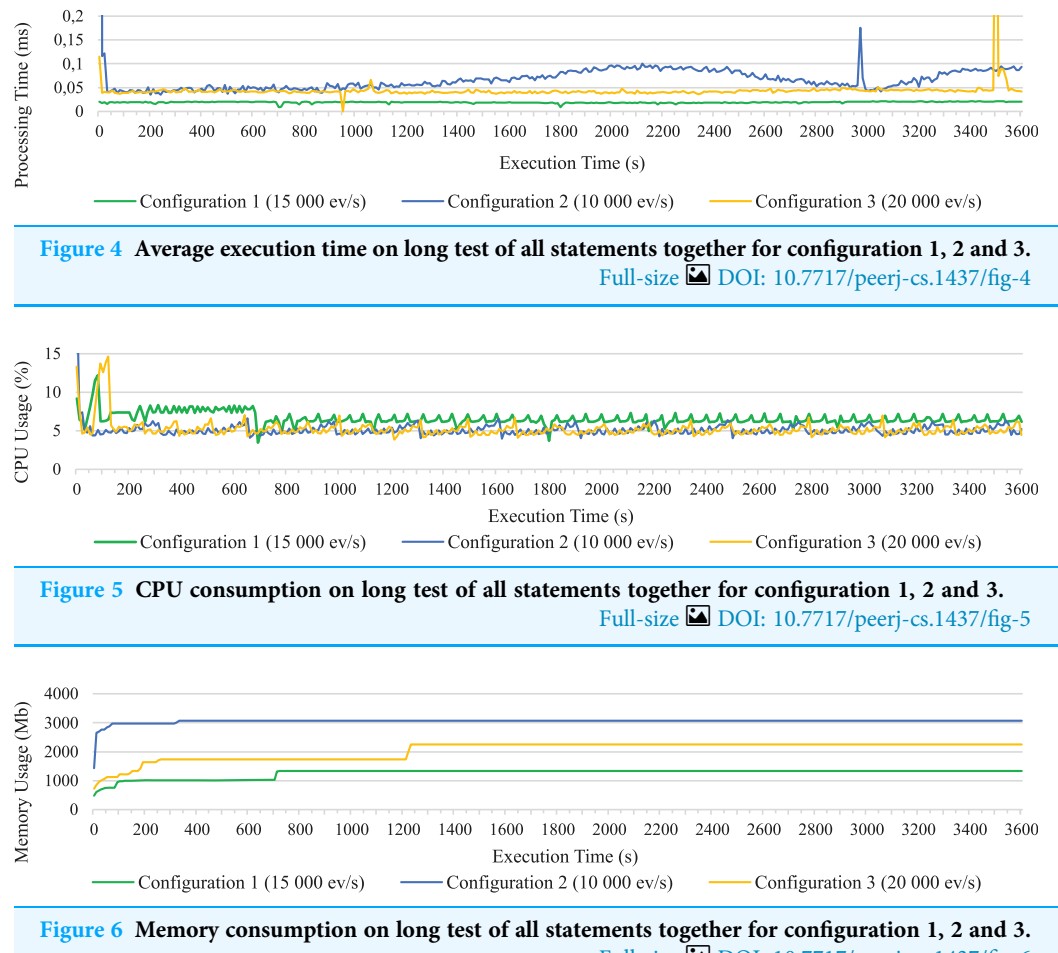

**Figure 4** Average execution time on long test of all statements together for configuration 1, 2 and 3.

**Figure 5** CPU consumption on long test of all statements together for configuration 1, 2 and 3.

**Figure 6** Memory consumption on long test of all statements together for configuration 1, 2 and 3.

The CPU average consumption of the system during the execution of the long test for the three configurations have also been represented in Fig. 5; representing only the CPU consumption of one of the machines in Configuration 3, since the values taken by both in the two distributed machines are similar. As we can see in the figure, the CPU consumption is very similar in the three configurations, and in all cases the consumption is not significant, below 10%.

Memory consumption of the system during the execution of the long test for the three configurations has also been represented in Fig. 6, respectively; representing only the memory consumption of one of the machines in configuration 3, since the values taken by both two distributed machines are similar. In Fig. 6 we can see how the open-source Esper option also presents the best values in terms of memory consumption. The graph remains very flat as it has been rounded around the GB as the values increase.

Finally, it is important to mention that we also tested the fault tolerance of the system with Esper HA; by shutting down one of the machines: no messages were lost, and the other machine continued to do all the work normally.

# RELATED WORK

Firstly, it is important to note that we found no directly related work that provides a performance comparison between different configurations of Esper's CEP engine with different brokers, but rather we have generally identified work that separately evaluates either several CEP engines, several message brokers, or the integration of a particular engine with a particular broker.

It is important to note that a number of benchmarks for IoT data processing platforms have been published in recent years, most notably (*Dayarathna & Suzumura, 2013*; *Shukla, Chaturvedi & Simmhan, 2017*; *Hesse et al., 2021*); however, they are not suited to the needs of the performance tests we propose in this article, as explained in the following lines. RioTBench (*Shukla, Chaturvedi & Simmhan, 2017*) classifies IoT tasks into parse, filter, statistical analytics, predictive analytics, pattern detection, visual analytics and IO operations. They propose a series of microbenchmarks for many of these categories, but none for pattern detection, which is precisely what CEP engines do and which, as they explain, could be embedded in the DSPS that they analyze. Regarding ESPBench (*Hesse et al., 2021*), again it should be noted that it is focused on DSPSs; besides, the system under test is only prepared for integration with Apache Kafka and limited, as indicated in the threads to validity section, to manufacturing applications. Although some of the queries could be reused for other domains by using a large number of core DSPS operations, we consider it preferable to create a more generic benchmark, which allows the separation of these operations. It is interesting the discussion of the results of Apache Spark in some cases (for example in the article query 1) which, although it can process the result more quickly by using independent micro-batches, it can lead to different results from those expected and, therefore, it is not comparable with the other systems in this type of scenario. Also the article (*Nasiri, Nasehi & Goudarzi, 2019*) is a relevant reference for the comparison of several DSPS; they compare the performance and scalability of Apache Storm, Apache Spark and Apache Flink, which provides FlinkCEP—a CEP library implemented on top of Flink, integrated with Kafka. They conclude that Flink behaves well at small-scale clusters, but it has poor scalability on the large-scale clusters. Again, there is no evaluation of various CEP operators. Finally, four scenarios with different stream processors (S, S4 and Esper) are evaluated in (*Dayarathna & Suzumura, 2013*). The results show a much better performance in Esper compared to the rest of the systems when a single node is used; the tests on distributed nodes (distributing the messages manually with the open-source version of Esper), show a worse performance in Esper, probably due, according to the authors, to delays in the communication network and serialization. This benchmark does not show which Esper operators have been included in the four scenarios and does not seem to include, for example, any time window, which is one of the key operators of Esper. On the contrary we have proposed a benchmark in which we see the use of several common operators in the use of Esper.

Regarding the benchmarking of various message brokers in various comparative analyses, we can see that Apache Kafka and RabbitMQ stand out from many others as fast and reliable brokers. However, there is no clear winner, as shown in the evaluation

presented in *Lazidis, Tsakos & Petrakis (2022)*, since Kakfa presents better throughput than RabbitMQ, while RabbitMQ presents a much lower latency. Therefore, it would be necessary to evaluate the needs of the particular case study throughput *vs* latency, when choosing one or the other option. Although in many cases Kafka can support very high input event rates, it should be noted that this depends on the case study. The article (*Langhi, Tommasini & Valle, 2020*) illustrates exactly that situation; in this article Kafka streams for complex event recognition is used and two examples are presented in the evaluation; while for one a throughput of 112,114 events per second is reached, for the second only a throughput of 2,565 events per second is achieved.

On the other hand, we found some comparisons between different rule/CEP engines; for example (*Rosa et al., 2015*) present a comparative study of correlation engines for security event management. Esper CEP engine is among the engines evaluated; in their analysis, we can see that the Esper engine has a very good performance with a high throughput, as well as a fast processing of large amounts of data. Although it is not the best under all circumstances, the authors consider it to be the most suitable in terms of an adequate compromise between performance, configuration flexibility and easiness of setup.

We also found an evaluation of the integration of the Esper engine with an ESB and the Mosquito broker (*Roldán et al., 2020*). There also exist some evaluations of the integration of Node-RED with Mosquito MQTT (*Kodali & Anjum, 2018*). Note that Node-RED provides a much more succinct syntax for defining event patterns and lower performance than the Esper CEP engine. Meanwhile, Mosquitto, a broker specifically designed for lightweight protocols for the IoT, does not reach the throughput and versatility of the RabbitMQ or Apache Kafka brokers.

In (*Aktas & Astekin, 2019*) several comparisons integrating various tools are also made. First, they compare three alternatives: Kafka plus Storm, Storm plus Esper and Esper plus Kafka, the latter being the one that obtains the best latency results up to an input rate of 50,000 events per second, taking into account that it is a simple test in which the data are processed without applying any processing pattern. The same tests are then performed for Kafka plus Spark, Spark plus Drools and Drools plus Kafka, with the last one obtaining the best results. Then they compare the integration of Apache Storm with Esper *vs* Apache Spark with Drools Fusion using a set of CEP rules; in this comparison we can see that the option including Esper got better results in the performance tests. The latest tests were only performed with a suitable latency for an incoming rate of 1,000 events per second, though.

Additionally, other evaluations of Esper CEP as found in the literature only focus on the integration of the open-source Esper CEP engine with a message broker. In the past, we proposed a microservice-based architecture in which one of the microservices is an Esper CEP engine (*Ortiz et al., 2022a*). Although the article performs a reasonably extensive evaluation, it proposes no benchmark, but rather evaluates the system with a event pattern linked to the case study by evaluating the time it takes to transfer events between microservices and to process them in the Esper microservice. Although we achieved higher input event rates than those achieved in this evaluation, it is due to the simplicity of the event pattern used (*select * from Dummy*) which did not require great memory or computation resources, the low number of complex events generated at the output and the

lack of sliding windows that overload the system. *Corral-Plaza et al. (2021)* evaluate how the integration of Esper with Kafka behaves with up to 32 partitions, processing up to 150,000 events per second. Again, the tests are performed without defining a benchmark, with a simple event pattern (*select* *); it is demonstrated that the system is highly scalable under these simple conditions. Also, in *Roldán-Gómez et al. (2021)*, the performance of Esper CEP integrated with Mule ESB is compared to the one of WSO2 Siddhi CEP engine and ESB (*WSO2, 2019*) in a network security scenario, without contemplating the chance of scaling through a distributed version of the CEP engines. Thus, all these works are valuable to see how Esper's open-source engine behaves in conditions where the event patterns do not demand a great amount of memory, which is what requires the greatest amount of system resources. It is also difficult to compare, for example, Esper's integration with Rabbit *vs* its integration with Kafka through these works, as they use different event patterns, machines and architectures; in any event, our work complements the information in these other publications.

Special attention deserves a article in which we evaluated and compared the open-source Esper CEP engine in an event-driven architecture with the use of an ESB compared to the use of data-flows both in a server and in a Raspberry Pi (*Ortiz et al., 2022b*). The benchmark used in this article is the one we extended for our tests here. Although our article focuses on processing on servers with good performance, other scenarios may require deployment on devices with fewer resources, such as a Raspberry Pi; article (*Ortiz et al., 2022b*) helps developers to see how the system would behave in this scenario.

To facilitate the understanding of this section, we have included Table 11, which summarizes each of the proposals discussed above. The table indicates for each work, the technology evaluated, the CEP operators evaluated in that work, the peak processing rate or maximum value of incoming messages (events) per second that the system is able to process and the limitations or disadvantages of this proposal in relation to the main objective of this article, which is to have a reference for the evaluation of complex event processing products, particularly in comparison with the open source and enterprise alternatives of Esper CEP. In relation to the peak processing rate, it has been determined to be the maximum processing rate as long as a latency of 1 s is not exceeded; this value has been shown for the operations or operators with which the lowest and highest ratio has been achieved.

Thus, to summarize this section, we can conclude that, to the best our knowledge, no work has compared the various CEP products offered by Espertech, nor has any work explicitly compared their integration with two of the most powerful message brokers on the market.

## Discussion and responses to research questions

In this section, we discuss the results of the tests and responses to the research questions, making special emphasis on responding to RQ3 to give a number of suggestions to be considered when implementing a new system based on a software architecture with Esper CEP.

**Table 11 Summary of related work reviewed.**

| Proposal | Technology evaluated | CEP operators evaluated | Peak processing rate (Messages per second) | Threats to validity/Drawbacks |
|---|---|---|---|---|
| DSPS | | | | |
| *Shukla, Chaturvedi & Simmhan (2017)* | Apache Storm for DSPS | None (IoT data processing operations evaluated). | 310 to 68,000 | No CEP operators evaluated. |
| *Hesse et al. (2021)* | Apache Kafka-Apache Flink for DSPS | None (Manufacturing operations evaluated). | 1,000 to 10,000 | Limited to Manufacturing Applications. No CEP operators evaluated. |
| | Apache Kafka-Hazelcast Jet for DSPS | | 1,000 to 10,000 | |
| *Nasiri, Nasehi & Goudarzi (2019)* | Apache Storm | None (Advertising application and model training application evaluated). 2 nodes. | 200 to 500,000 | No CEP operators evaluated. |
| | Apache Storm-No Ack | | 200 to 500,000 | |
| | Apache Flynk | | 100 to 500,000 | |
| | Spark Streaming | | 150 to 400,000 | |
| *Dayarathna & Suzumura (2013)* | S | Not described (microbenchmark). | 10,000 | CEP operators evaluated are not described. |
| | S4 | | 3,000 | |
| | Esper | | 10,000 | |
| Message brokers | | | | |
| *Lazidis, Tsakos & Petrakis (2022)* | Apache Kafka | None (Processing 100 Bytes message). | 80,436 | No CEP operators evaluated. |
| | RabbitMQ | | 61,824 | |
| Complex event processing | | | | |
| *Langhi, Tommasini & Valle (2020)* | Kafka Streams Processor | (a) Every A followed by B (b) Every A followed by every B. | (a) 112,114 (b) 2,565 | Only 1 CEP operator evaluated (followed by). |
| *Rosa et al. (2015)* | Esper | 20 event patterns (operators not specified). | 38,461 | CEP operators evaluated are not described. |
| | Drools | | 21,272 | |
| | NodeBrain | | 6,369 | |
| | SEC | | 4,405 | |
| *Roldán et al. (2020)* | Mosquito-MULE ESB-Esper | Time to detect network security patterns evaluated. | No performance evaluation. | No CEP operators evaluated. |
| *Kodali & Anjum (2018)* | NodeRed-Mosquitto | | No performance evaluation | No CEP operators evaluated. |
| *Aktas & Astekin (2019)* | Kafka-Storm | Six event patterns (arithmetic comparison operators and time windows). | 5,000 | Suitable latency is only guaranteed up to 1,000 message/second rate. Only open source Esper engine is evaluated. |
| | Storm-Esper | | 50,000 | |
| | Esper-Kafka | | 50,000 | |
| | Kafka-Sparks | | 1,000 | |
| | Spark-Drools | | 50,000 | |
| | Drools-Kafka | | 50,000 | |
| *Ortiz et al. (2022a)* | RabbitMQ-Esper | Three event patterns (statistic operations, comparisons, grouping and time windows). | 50,000 | Low number of even patterns. Only open source Esper engine is evaluated. |

(Continued)

| Table 11 (continued) | | | | |
|---|---|---|---|---|
| Proposal | Technology evaluated | CEP operators evaluated | Peak processing rate (Messages per second) | Threats to validity/Drawbacks |
| *Corral-Plaza et al. (2021)* | Kafka-Esper | One dummy event pattern (select *). | 150,000 (eight partitions) | Very simple pattern evaluated. Only open source Esper engine is evaluated. |
| *Roldán-Gómez et al. (2021)* | MQTT-Mule ESB-Esper | Time to detect network security patterns compared. | No performance evaluation. | No CEP operators evaluated. |
| | MQTT-WS02 ESB-Siddhi | | | |
| *Ortiz et al. (2022b)* | RabbitMQ-Mule ESB-Esper | Six event patterns (statistic operations, comparisons, grouping and time windows). | 5,000 to 10,000 | Only open source Esper engine is evaluated. |
| | RabbitMQ-Esper Dataflows | | 10,000 | |

# DISCUSSION

Prior to discussion, it is worth emphasizing that the Esper engine was chosen, as mentioned, because of its good reputation for robustness and performance, as well as its very extensive grammar, which allows a wide variety of event patterns and functionalities to be defined in an intuitive language, similar to SQL. Besides, we have had experiences of real use cases with companies in our environment in which the companies have opted for the use of Esper, particularly with the port authority of the Bay of Cadiz and the Puerto Real Energy Group (GEN) in the areas of air quality control (*Ortiz et al., 2022a*) and water supply network management (*Corral-Plaza et al., 2020*), respectively. We are currently working on a joint project with the company GEN that has led us to carry out this study of the various options of Esper CEP and its integration with RabbitMQ and Kafka to see which are the most convenient options for them.

One of the most striking results is that in the case of working with a centralized architecture, Esper's open-source engine performs very well; when engaging in centralized processing on a single machine, the system performs better when we integrate Esper CEP with RabbitMQ, even in its open-source version, as the system is not overloaded as much with Kafka and Esper Enterprise-HA-EQC backups. In general, we find a poorer average processing time with the Esper Enterprise-HA-EQC version compared to the open-source one. The reason for the lower performance may be the backups that Esper HA makes to be able to recover the system after a failure. However, in both implementations, the tests show similar memory and CPU consumption, although, in general, it is slightly better for the open-source version.

However, when the event load is kept high over time, the integration of open-source Esper with RabbitMQ was unable to handle a continuous load of 20,000 events per second. However, the CEP engine is capable of processing such a load, handling it at the broker and sending it over the network overloaded the system. It is worth noting that we are unlikely to have a system that receives such a rate of data input over a sustained period of time and

that, where appropriate, more instances of the broker in question could be deployed and have multiple incoming queues to the CEP engine.

It is also important to remember that, although with distributed processing we can obviously achieve better performance, especially in long tests, distribution is only possible in systems where the processing done on one machine does not depend on another, as there is no communication between the different distributed CEP installations: the different engines cannot share information with each other to apply a event pattern to shared information. That is, for the distributed architecture to be useful, we must apply it to an environment in which all the nodes can independently apply the operations of the event patterns on the dataset they receive, without being able to relate the complex events of the different engines.

Moreover, it is also clear that the system consumes much more memory resources with time windows than with data windows; it is key to take this issue into account when deciding which event patterns to deploy in our system.

## Responses to research questions

Response to RQ1. In a centralized architecture, with a single Esper CEP engine, what are the advantages and disadvantages of integrating it with two competing brokers such as RabbitMQ and Kafka and which one should be used to achieve the best performance in real-time stream data processing?

Based on the previous discussion, in a centralized architecture with a single Esper CEP engine, if we do not need to have a system backup, the open-source version of Esper integrated with RabbitMQ provides better performance compared to its integration with Kafka. Although the memory and CPU consumptions are similar in both cases, we get better latency and throughput time in its integration with Rabbit. However, if you have high reliability requirements and need a backup, you will be forced to use the Esper Enterprise-HA-EQC version integrated with Kafka. In short, RabbitMQ offers the advantage of better latency and time throughput; whereas Kafka will offer reliability and, in case you need rates above 10,000 or 15,000 events per second, you could do it through the horizontal scalability options of Esper Enterprise-HA-EQC.

Response to RQ2. When does it outweigh using a distributed CEP architecture to achieve greater horizontal scalability and how does this impact system performance?

Depending on the patterns we have deployed in the system, from 15,000 or 20,000 input events per second, it may be necessary to make use of a distributed architecture that allows us to scale CEP horizontally. This horizontal scaling will imply worse performance rates than a centralized architecture when we have lower input event rates, especially in terms of memory consumption, but without limiting the good performance of the system; but it is definitely the only option to achieve good, sustained performance with constant high event rates over time. It is important to stress the limitation of Esper CEP with Kafka for horizontal scalability: there can be no dependencies between the events processed on the distributed machines nor between the complex events detected on these, since there is no communication between the distributed engines.

Response to RQ3. Which of these Esper engines and which messaging broker should I use depending on my system requirements?

In light of the above discussion and the answer to previous RQs, we provide the following recommendations, including some limitations:

- If there is no compelling need for fault tolerance and our system is going to do centralized processing on a single machine, we suggest using free and open-source software for the implementation of our architecture; in particular, the open-source Esper CEP engine and RabbitMQ.
- If we need high availability and a higher fault tolerance, we must use Esper HA, but being aware that the integration has to be done with Kafka.
- The same applies to horizontal scalability, if our domain has no dependencies between complex events or between simple events of different types, we can scale the system using an Enterprise Esper CEP configuration with EQC. However, in the case of having dependencies between them, we will be unable to scale using the distributed option, but will need a more powerful machine and this is the main limitation of the horizontal scalability with Esper CEP and Kafka.
- In terms of choosing RabbitMQ over Apache Kafka, in the architectures to be used should take into account that RabbitMQ provides better latency, especially for low workloads, but Apache Kafka deals better with higher workloads. Therefore, in case of having very high workloads, but no need for very low latency, it is better to use Apache Kafka. However, if low latency is needed and workloads are not so high over a long period of time, it is advisable to go for RabbitMQ.
- Besides, as for the definition of the event patterns, it is the time windows that most overloads the system. If we can implement the same functionality with another type of operator, our system will probably consume much less memory. So, if we intend to use time windows, we need a computer with high RAM availability.

We consider this study to be applicable to multiple contexts and application domains related to the processing and correlation of real-time data from IoT or smart city environments. As previously introduced, Esper CEP's EPL language provides a wide grammar that provides a great versatility for the definition of the patterns to be detected; this added to its good performance and native integration with some message brokers such as RabbitMQ and Kafka, postulate it as a suitable candidate for these scenarios. However, each scenario has its own particular characteristics and may require the use of operators other than those used in the benchmark proposed in this article, but I think that the study can be extrapolated to other operators in terms of memory or CPU consumption and scalability. However, we must always bear in mind the limitation of a scenario in which we have a very high load of incoming events per second maintained over time and where there are many dependencies between incoming events to the system that hinder horizontal scalability, in which case it may require a greater effort in the design of the topics used in the message broker and the way of distributing the events between different machines for processing.

## CONCLUSIONS

In conclusion, we can say that the different options offered by Esper for the CEP provide highly efficient solutions for real-time data processing. There is a wide range of options, from the free and open-source version for centralized processing, without fault tolerance but with direct integration with different message brokers, to paid products with fault tolerance availability and high horizontal scalability, although with limitations in terms of the brokers to be used for data integration. This article complements other evaluation articles that focus on other aspects of DSPs and provides additional tools for choosing a CEP engine and messaging broker for real-time data processing and correlation.

In our future work, we will perform the tests within a company setting with real data and event patterns tailored to the company's needs. This is a water supply management company that aims to detect fraud and leaks in the city's water network in real time. It is therefore a scenario where there is no great need for fault tolerance since the fact that data is lost for a few minutes does not have a serious impact on the detection of fraud or leaks. In order to save costs for the company, without reducing efficiency and given that there is no critical need for fault tolerance, our best proposal is to implement the system with the open-source version Esper's CEP engine. Once we test the system with real data and custom event patterns, we expect the system to perform satisfactorily. In the event of the need to scale horizontally, the city could be divided into sectors and the data from each sector to different machines, where the same event patterns would be applied on different data sets using Esper Enterprise-HA-EQC version, although, in this case, with a higher economic cost for the company. While the company has identified the situations they wish to detect and we have manually defined the event patterns according to the corresponding simple event correlation in each case, we are also planning a new collaboration in which we would use machine learning techniques to learn new event patterns for their domain.

## ACKNOWLEDGEMENTS

We would like to thank GEN Grupo Energético for their willingness to share their data for future tests to be carried out for the management of water supply networks in the area of the city of Puerto Real.

### Funding

This work was supported by grant programme for R&D&I projects, for universities and public research entities qualified as agents of the Andalusian Knowledge System, within the scope of the Andalusian Plan for Research, Development and Innovation (PAIDI 2020). Project 80% co-financed by the European Union, within the framework of the Andalusia ERDF Operational Programme 2014–2020 "Smart growth: an economy based on knowledge and innovation". Project funded by the Ministry of Economic Transformation, Industry, Knowledge and Universities of the Andalusian Regional Government. The DECISION project with reference P20_00865, the AwESOMe grant PID2021-122215NB-C33 funded by MCIN/AEI/10.13039/501100011033/ and by ERDF A way to do Europe

and ASSENTER Grant PDC2022-133522-I00 funded by MCIN/AEI/10.13039/501100011033 and by the European Union Next Generation EU/PRTR. There was no additional external funding received for this study. The funders had no role in study design, data collection and analysis, decision to publish, or preparation of the manuscript.

**Grant Disclosures**

The following grant information was disclosed by the authors:
R&D&I Projects, Andalusian Knowledge System, Andalusian Plan for Research, Development and Innovation: PAIDI 2020.
Project 80% Co-financed by the European Union, Andalusia ERDF Operational Programme 2014-2020.
Ministry of Economic Transformation, Industry, Knowledge and Universities of the Andalusian Regional Government.
DECISION Project with Reference: P20_00865.
AwESOMe Grant PID2021-122215NB-C33: MCIN/AEI/10.13039/501100011033/.
ERDF A way to do Europe and ASSENTER Grant PDC2022-133522-I00: MCIN/AEI/10.13039/501100011033.
European Union Next Generation EU/PRTR.

**Competing Interests**

The authors declare that they have no competing interests.

**Author Contributions**

- Guadalupe Ortiz conceived and designed the experiments, analyzed the data, prepared figures and/or tables, authored or reviewed drafts of the article, and approved the final draft.
- Adrian Bazan-Muñoz performed the experiments, analyzed the data, performed the computation work, authored or reviewed drafts of the article, and approved the final draft.
- Winfried Lamersdorf conceived and designed the experiments, authored or reviewed drafts of the article, and approved the final draft.
- Alfonso Garcia-de-Prado performed the experiments, analyzed the data, performed the computation work, prepared figures and/or tables, and approved the final draft.

**Data Availability**

The raw data is available in the Supplemental Files and at Mendeley: Ortiz, Guadalupe; Bazán-Muñoz, Adrián; Lamersdorf, Winfried; Garcia-de-Prado, Alfonso (2023), "Evaluating the integration of complex event processing and message brokers", Mendeley Data, V1, DOI: 10.17632/ny5j6hm6g2.1.

**Supplemental Information**

Supplemental information for this article can be found online at http://dx.doi.org/10.7717/peerj-cs.1437#supplemental-information.

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
