# Peer review of "Evaluating the integration of Esper complex event processing engine and message brokers"

_PeerJ Computer Science, doi:10.7717/peerj-cs.1437_

## Round 0.1 · original submission · Minor Revisions

The authors should consider the comments of reviewer 2 to be considered for publication in PeerJ Computer Science.

·

Basic reporting

This is second review of the paper. Paper was substantially improved based on reviewer´s comments. All comments were addressed.

Experimental design

Reviewer´s comments were addressed:
Aim of the paper was explicitly defined in the Introduction. Research questions were defined.
Materials and Methods section was added.

Validity of the findings

Reviewer´s comments were addressed: research gap was identified.

Additional comments

Reviewer´s comments were addressed.

·

Basic reporting

This submission presents a work to integrate Esper complex event processing engine and some message brokers. The paper introduces the problem, provides a background and motivation, and presents the implementation of the software architectures to be evaluated, followed by a description of the hardware resources used and the methods followed. After that, the authors present an empirical evaluation, a related work, and a discussion of the obtained results.

The paper is worth reading and well-referenced.
The authors provide a detailed background.
The different figures and tables presented by the authors are relevant to the content of the paper.

Experimental design

The authors have presented concise and relevant research questions. And the rest of the article heads in the right direction to provide answers to these questions.

Validity of the findings

The topic addressed in this paper is important, and the obtained results are interesting. Besides, the paper clearly defends the novelty and the interest of this proposed integration.


It will be interesting to see the application of this approach in a real case study, as the authors mentioned in the conclusion.

Additional comments

The authors provide an implementation of the software architecture and also discuss the obtained results. These two parts represent the core of their paper. However, I have a few additional comments:

- The authors explain why Esper was chosen as the "most appropriate" Event processing engine. However, I would appreciate seeing a detailed benchmark (in table format with the important comparative criteria), comparing Esper to other CEP engines, such as Apache Flink, Sidhi, etc, to corroborate their choice. This table will summarize the content of the "related work" section and also makes it easier to read.

- CEP rules have not been sufficiently discussed in this paper. How will the authors define these rules that are going to be used in the CEP engine? Are the rules defined manually or do they use (or intend to use) a method to automatically learn CEP rules?

- I have also a comment regarding the distributed CEP implementation. What about the cost? What do you propose to reduce the communication costs of distributed event processing? The other measurements presented in this paper are important but the cost is a non-negligible measurement in this kind of case.

---

## Round 0.2 · Minor Revisions

Please ensure that the statistical analysis included in the article has been performed according to the technical standard required for publication. In particular, sample size, appropriate statistical measures, correction for multiple testing, effect size, degrees of freedom, and report relevant figures such as the statistic, n and exact p-values/confidence intervals as required.

·

Basic reporting

This revision has significantly improved the paper. The authors have adequately addressed the comments included in my first review. Therefore, I have no further comments, and I would recommend the paper for publication.

Experimental design

No further comments.

Validity of the findings

No further comments.

Additional comments

No additional comments.

---

## Round 0.3 · accepted · Accept

Congratulations! Your article is ready to be published in the journal PeerJ Computer Science. We appreciate your interest in publishing your work in our journal and look forward to receiving further quality work from you.